# I$^2$EBench: A Comprehensive Benchmark for Instruction-based Image Editing

**Yiwei Ma**[1*]    **Jiayi Ji**[1*]    **Ke Ye**[1]    **Weihuang Lin**[1]    **Zhibin Wang**[2†]
**Yonghan Zheng**[1]    **Qiang Zhou**[2]    **Xiaoshuai Sun**[1†]    **Rongrong Ji**[1]
[1] Key Laboratory of Multimedia Trusted Perception and Efficient Computing,
Ministry of Education of China, Xiamen University, 361005, P.R. China.
[2] Inf Tech Company, Hangzhou, 310000, P.R. China.
yiweima@stu.xmu.edu.cn    xssun@xmu.edu.cn

## Abstract

Significant progress has been made in the field of Instruction-based Image Editing (IIE). However, evaluating these models poses a significant challenge. A crucial requirement in this field is the establishment of a comprehensive evaluation benchmark for accurately assessing editing results and providing valuable insights for its further development. In response to this need, we propose **I$^2$EBench**, a comprehensive benchmark designed to automatically evaluate the quality of edited images produced by IIE models from multiple dimensions. I$^2$EBench consists of 2,000+ images for editing, along with 4,000+ corresponding original and diverse instructions. It offers three distinctive characteristics: 1) *Comprehensive Evaluation Dimensions:* I$^2$EBench comprises 16 evaluation dimensions that cover both high-level and low-level aspects, providing a comprehensive assessment of each IIE model. 2) *Human Perception Alignment:* To ensure the alignment of our benchmark with human perception, we conducted an extensive user study for each evaluation dimension. 3) *Valuable Research Insights:* By analyzing the advantages and disadvantages of existing IIE models across the 16 dimensions, we offer valuable research insights to guide future development in the field. We will open-source I$^2$EBench, including all instructions, input images, human annotations, edited images from all evaluated methods, and a simple script for evaluating the results from new IIE models. The code, dataset and generated images from all IIE models are provided in github: https://github.com/cocoshe/I2EBench.

## 1    Introduction

Instruction-based Image Editing (IIE)Brooks et al. [2023], Geng et al. [2023], Zhang et al. [2024a], Li et al. [2023c], Wang et al. [2023b], Zhang et al. [2023a], Fu et al. [2024], which aims to edit an image using a text instruction, provides a user-friendly way for the community to edit images. Over the past few years, significant progress has been made in IIE, with the development of diffusion models Ho et al. [2020], Sohl-Dickstein et al. [2015], Welling and Teh [2011], Kulikov et al. [2023] and large vision-language models (LVLMs) Liu et al. [2023a,b], Fei et al. [2024c,a,b], Ma et al. [2024]. However, there is a pressing need for a comprehensive benchmark to effectively assess the performance of these models. An ideal evaluation framework should not only measure the editing quality across different dimensions but also align with human perception to ensure reliable measurements. Furthermore, the evaluation should highlight the specific strengths and weaknesses of each model, thereby offering valuable insights for future endeavors in data selection, training

---

[*]Equal contribution.
[†]Corresponding author.

38th Conference on Neural Information Processing Systems (NeurIPS 2024).

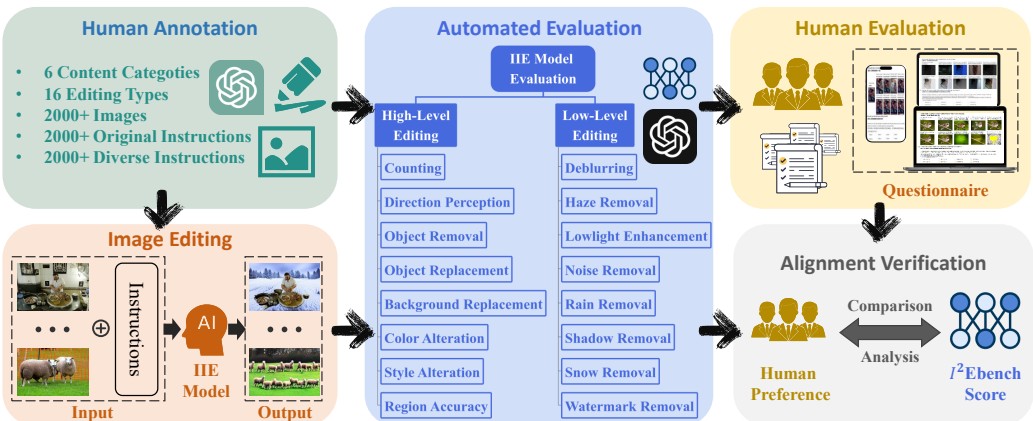

Figure 1: Overview of I²EBench, an automated system for evaluating the quality of editing results generated by instruction-based image editing (IIE) models. We collected a dataset of over 2000+ images from public datasets Lin et al. [2014], Guo et al. [2023b], Martin et al. [2001], Chen et al. [2021], Ancuti et al. [2019], Liu et al. [2021b,a], Qu et al. [2017], Nah et al. [2017], Shen et al. [2019], Wei et al. [2018] and annotated them with corresponding original editing instructions. To diversify the instructions, we used ChatGPT Achiam et al. [2023] to generate varied versions. With the collected images and the original/diverse editing instructions, we utilized existing IIE models to generate edited images. Subsequently, we developed an evaluation methodology to automatically assess the adherence of edited images to the provided instructions under different dimensions. We also implemented human evaluation to obtain human preferences for editing results of different IIE models. Finally, we analyzed the correlation between automated evaluation and human evaluation, confirming alignment with human perception.

strategy selection, and architecture design within this field. However, evaluating an IIE model poses challenges due to the diverse range of editing types and the inherent difficulty in assessing the level of alignment between edited images and given instructions.

Existing evaluation metrics for IIE could be divided into three categories: 1) conventional metric; 2) user study; 3) benchmark. The first category Brooks et al. [2023], Geng et al. [2023], Zhang et al. [2024a], Li et al. [2023c], Wang et al. [2023b], Huang et al. [2024b] employs conventional metrics to evaluate IIE models, including CLIP Score Radford et al. [2021], CLIP Text-Image Direction Similarity Radford et al. [2021], PSNR Korhonen and You [2012], SSIM Wang et al. [2004], and LPIPS Zhang et al. [2018]. The advantage of this approach is its ease of use. However, a single metric is not suitable for evaluating all types of editing. For instance, CLIP score measures the similarity between images and text, making it less suitable for low-level visual editing tasks like denoising and low-light enhancement. Similarly, PSNR, which measures image similarity, is not adequate for high-level visual editing tasks such as object removal and replacement. The second category Li et al. [2023c], Zhang et al. [2023a], Fu et al. [2024] involves methods that evaluate the effectiveness of different techniques by soliciting ratings from human participants. This approach directly reflects human preferences and aligns the results with human perception. However, it is a costly method and lacks reproducibility, as the test sets and participants may be not consistent in each evaluation. The final category comprises benchmarks Kawar et al. [2023], Wang et al. [2023c], Basu et al. [2023], Huang et al. [2024a] specifically designed for evaluating IIE models. While these benchmarks are tailored for IIE, they have certain limitations. For example, TedBench Kawar et al. [2023] evaluates only 100 images with commonly occurring editing types, which may not sufficiently demonstrate the capabilities of IIE models. EditBench Wang et al. [2023c] focuses on mask-guided editing, rendering it unsuitable for evaluating mask-free methods. In EditVal Basu et al. [2023], only a limited set of dimensions related to size or location can be automatically evaluated, limiting its universality.

In this paper, we propose I²EBench, a comprehensive benchmark designed to automatically evaluate the performance of IIE models. I²EBench exhibits three attractive characteristics: 1) Comprehensive Evaluation Dimension, 2) Human Perception Alignment, and 3) Valuable Research Insights.

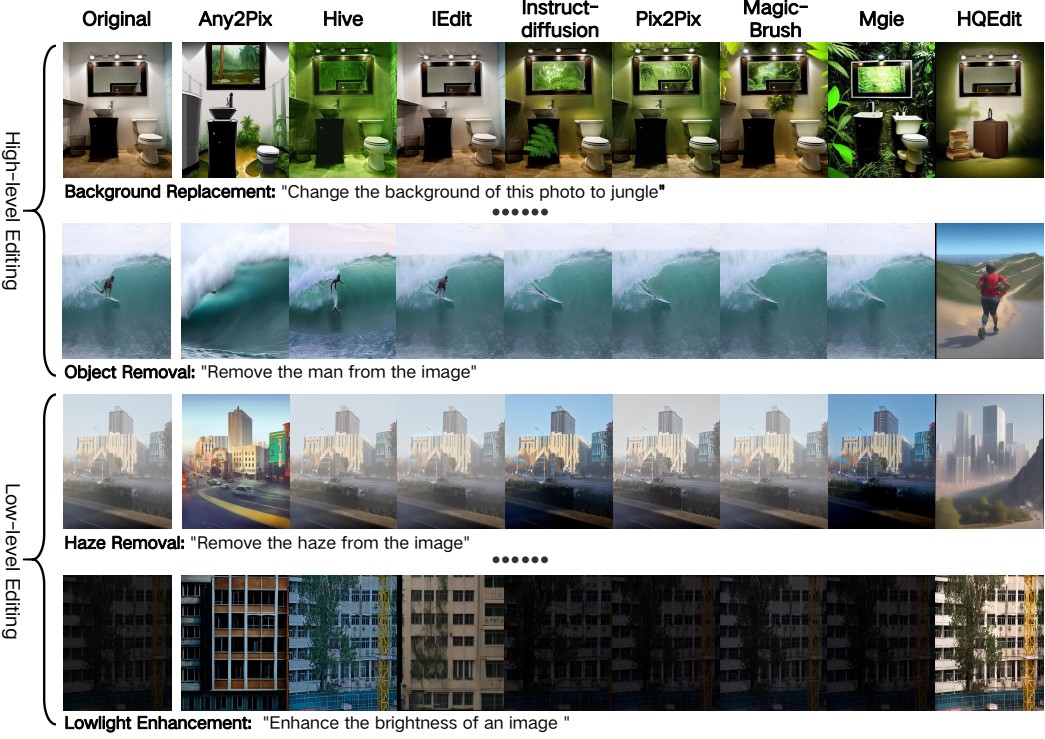

Figure 2: Visualization of the editing results on the proposed 16 evaluation dimensions using different IIE models, including InstructAny2Pix Li et al. [2023c], HIVE Zhang et al. [2023a], InstructEdit Wang et al. [2023b], InstructDiffusion Geng et al. [2023], InstructPix2Pix Brooks et al. [2023], MagicBrush Zhang et al. [2024a], MGIE Fu et al. [2024], and HQEdit Hui et al. [2024]. A detailed version can be found in supplementary materials.

First and foremost, I$^2$EBench offers a comprehensive evaluation dimension. These dimensions are categorized into two main types: *High-level Editing* and *Low-level Editing*. High-level editing primarily focuses on understanding instructions or editing specific areas of images, whereas low-level editing is more concerned with editing image details or the entire image. As shown in Fig. 1, both high-level and low-level editing consist of 8 fine-grained editing dimensions, which serve to demonstrate the model's proficiency in high-level and low-level editing. We meticulously collected approximately 140 images for each editing dimension and annotated each image with an original editing text instruction. To diversify the instructions, we also utilized ChatGPT Achiam et al. [2023] to enhance the description of the instructions and obtain a wider range of variations. In addition to the multi-dimensional evaluation, we also conducted a multi-category evaluation to assess the model's performance on different content categories. To achieve this, we included additional annotations for each instruction with different categories such as Animal, Object, Scenery, Plant, Human, and Global.

Second, the I$^2$EBench score aligns with human perception. This is accomplished by collecting scores from human annotators for the outputs generated by different IIE models, covering multiple evaluation dimensions. By conducting a comprehensive analysis of both the I$^2$EBench scores and the human scores, we have identified a substantial correlation between them. This discovery serves as compelling evidence, affirming that our proposed evaluation approach closely aligns with human perception.

Lastly, I$^2$EBench offers valuable research insights through its systematic evaluation across various dimensions and categories. The proposed I$^2$EBench not only facilitates a comprehensive assessment of existing models but also derives valuable insights into their respective strengths and weaknesses. These insights act as a roadmap for enhancing architecture design, refining data selection strategies, and ultimately elevating the quality of editing outcomes.

We are open-sourcing I$^2$EBench, including all instructions, input images, human annotations, edited images from all evaluated methods (like Fig. 2), and a simple script for evaluating the results of new IIE models. By making these resources freely available, we aim to foster fair comparisons within the field and facilitate valuable insights for community development.

## 2 Related Work

### 2.1 Instruction-based Image Editing

With the advancements in Generative Adversarial Networks (GAN)Goodfellow et al. [2014, 2020], Mao et al. [2017], Karras et al. [2019], Yoon et al. [2019], Karras et al. [2020a], Chen et al. [2018], Zhang et al. [2019] and Diffusion modelsSong et al. [2020], Ho et al. [2020], Nichol and Dhariwal [2021], Kawar et al. [2022], Austin et al. [2021], Dockhorn et al. [2022], text-to-image models Saharia et al. [2022], Rombach et al. [2022], Ramesh et al. [2021, 2022], Betker et al. [2023], Karras et al. [2019, 2020b] have made remarkable progress in recent years. As the demand for image editing continues to grow, a multitude of text-based image editing Xu et al. [2024], Kawar et al. [2023], Zhang et al. [2023b], Saund et al. [2003], Zhang et al. [2024b] models have emerged. One editing task, known as Prompt-based Image Editing (PIE) Avrahami et al. [2022], Valevski et al. [2023], Hertz et al. [2022], Dong et al. [2023], requires users to provide a target description along with the original image. The PIE model then analyzes the target description to modify the input image accordingly, generating a target image that matches the provided description. However, despite the lowered threshold for image editing, the requirement of describing the entire content of the target image in the description still poses challenges in terms of user interaction. To address this limitation, Instruction-based Image Editing (IIE) Brooks et al. [2023], Geng et al. [2023], Li et al. [2023c], Fu et al. [2024], Huang et al. [2024b] was proposed, which simplifies the user's role to providing the original image and modification instructions (*e.g.,* 'Remove the dog'). One notable implementation, InstructPix2Pix Brooks et al. [2023], introduces a large-scale dataset for instruction-based image editing. The dataset is created using a fine-tuned GPT-3 Brown et al. [2020] and image pairs generated by the Prompt-to-Prompt diffusion model Hertz et al. [2022]. Additionally, InstructPix2Pix proposes an instruction-based diffusion model for image editing based on this dataset. However, due to the automatic generation and filtering of the InstructPix2Pix dataset, concerns arise regarding its quality and potential noise. To address this, MagicBrush Zhang et al. [2024a] proposes a manually-annotated instruction-guided image editing dataset. In addition to textual instructions, InstructAny2Pix Li et al. [2023c] proposes a model that utilizes other modalities, such as audio and image, as instructions. To enhance the level of detail in instructions and improve the accuracy of editing results, MGIE Fu et al. [2024] introduces the use of Multimodal Large Language Models (MLLM) Liu et al. [2023a]. SmartEdit Achiam et al. [2023], aiming to improve the editing capabilities of IIE models in complex scenes, incorporates MLLM into the IIE model to better comprehend instructions. Despite significant progress, evaluating the editing performance of IIE models remains a crucial concern. Therefore, in this paper, we present I$^2$EBench, a systematic evaluation framework for these models. Our work includes an in-depth analysis of their strengths and weaknesses, offering valuable insights for the future development of IIE models.

### 2.2 Text-based Image Editing Benchmark

While numerous benchmarks Marino et al. [2019], Hudson and Manning [2019], Bigham et al. [2010], Lu et al. [2022], Li et al. [2023d,a,b], Yu et al. [2023], Wu et al. [2023b] have been introduced for evaluating vision-language tasks Wang et al. [2024b,a, 2022], Wu et al. [2023a], Dai et al. [2024], Hu et al. [2024], the evaluation of text-based image editing models often relies on metrics such as CLIP Score Radford et al. [2021], PSNR Korhonen and You [2012], SSIM Wang et al. [2004], and LPIPS Zhang et al. [2018]. Several existing studies have introduced benchmarks to assess the performance of image editing models. TedBench Kawar et al. [2023] presents a relatively small benchmark consisting of only 100 images and a limited set of highly common editing types. EditBench Wang et al. [2023c] is specifically designed to evaluate mask-guided image editing methods, which necessitate the availability of additional masks indicating the areas to be edited. In EditVal Basu et al. [2023], the evaluation of certain dimensions relies on manual labor, thereby limiting the reproducibility of performance. Moreover, the remaining dimensions primarily involve modifications to object size or position, lacking comprehensive coverage. While MagicBrush Zhang et al. [2024a] and Emu Edit Sheynin et al. [2023] propose test sets for evaluating editing performance,

they still rely on conventional metrics such as L1, L2, CLIP-I, DINO, and CLIP-T, which may not accurately capture the nuances of all editing types. SmartEdit Huang et al. [2024b] specifically develops a benchmark tailored for complex editing scenarios, but it does not accommodate other editing scenarios. Considering the current absence of a systematic benchmark that comprehensively evaluates the editing performance of IIE models across different editing types, we propose I$^2$EBench to address this gap.

# 3  I$^2$EBench

This section provides an overview of the main components of I$^2$EBench. In Sec. 3.1, we provide a concise introduction to the principles, definitions, and evaluation methods of 16 dimensions. Sec. 3.2 outlines the process of data annotation. Lastly, in Sec. 3.3, we present the human evaluation process to assess the correlation between the I$^2$EBench score and the human score. *A detailed explanation can be found in the supplementary materials.*

## 3.1  Evaluation Dimension

In our evaluation of the IIE model's editing quality, we have categorized it into 16 dimensions, each assessing different aspects of editing in a top-down manner. An overview of I$^2$EBench is presented in Fig. 1. High-level Editing Evaluation primarily focuses on assessing the model's ability to accurately understand instructions and make precise edits to local areas of the input image. This evaluation consists of 8 dimensions. Low-level Editing Evaluation, on the other hand, primarily examines global editing and detailed image processing. It also comprises 8 evaluation dimensions. Unlike previous approaches Fu et al. [2024], Zhang et al. [2023a], Geng et al. [2023] that relied on a single metric, such as CLIP score Radford et al. [2021], to evaluate editing quality for all editing types, we have developed specialized evaluation methods for each of the 16 dimensions. This approach is necessary due to the distinct goals of high-level and low-level editing.

### 3.1.1  High-level Editing

Evaluating editing quality in high-level dimensions poses a challenge due to the diverse goals involved, making it impractical to rely on a single metric. The advancement of Multimodal Large Language Models (MLLM) Gao et al. [2024], Chu et al. [2024], Zhu et al. [2024], Dong et al. [2024], Ma et al. [2022, 2023], Ji et al. [2022], such as GPT-4V Achiam et al. [2023], Gemini Pro Reid et al. [2024], and QWen-VL-Plus Bai et al. [2023], has significantly enhanced automated understanding of images. Therefore, to ensure precise evaluation of the editing quality of IIE models in high-level dimensions, we leverage the exceptional capabilities of the widely recognized GPT-4V model to make judgments for most high-level evaluation dimensions.

**Counting.** The Counting dimension pertains to instructions related to the number of objects, such as "add two apples to the image." To assess this dimension, we query GPT-4V about the number of target objects in the image and compare its response with the human-annotated answer.

**Direction Perception.** The Direction Perception dimension requires the IIE model to comprehend directions provided in instructions, and accurately make edits when presented with images. We evaluate this dimension by asking GPT-4V if the target object is in the expected position.

**Object Removal.** The Object Removal dimension focuses on removing the target object according to the given instruction. To evaluate this dimension, we inquire whether GPT-4V identifies the presence of the target object in the image.

**Object Replacement.** The Object Replacement dimension aims to replace the original object with the target object as instructed. To assess this dimension, we query GPT-4V about the presence of the target object in the image.

**Background Replacement.** The Background Replacement dimension involves replacing the original background with the target background as specified in the instruction. To evaluate this dimension, we ask GPT-4V if the background of the image matches the textual instruction.

**Color Alteration.** In the Color Alteration dimension, we modify the color of the target object using instructions. To evaluate this dimension, we inquire GPT-4V about the color of the target object in the edited image.

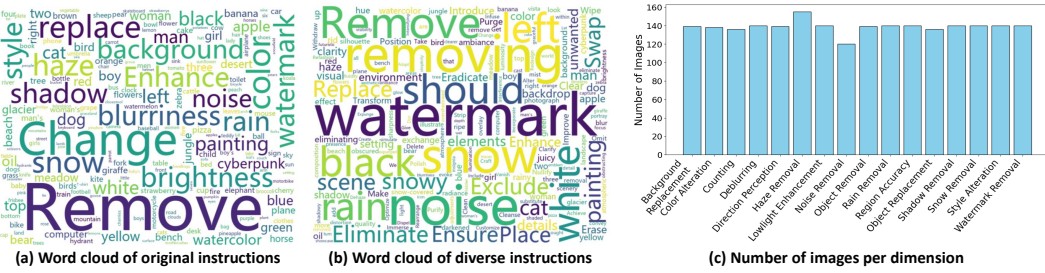

| (a) Word cloud of original instructions | (b) Word cloud of diverse instructions | (c) Number of images per dimension |

Figure 3: Word cloud visualization (a,b) and image quantity statistics (c) of I²EBench.

**Style Alteration.** The Style Alteration dimension focuses on changing the style of the image. To evaluate this dimension, we calculate the CLIP similarity Radford et al. [2021] between the edited image and "an image with ⋯ style".

**Region Accuracy.** In the editing task, we not only assess whether the target area has been edited correctly but also whether areas that should not be edited have been altered. To evaluate this dimension, we sample input images and instructions from the Object Removal, Object Replacement, and Color Alteration dimensions. We annotate the mask for the area that requires editing. Next, we fill the mask area of the images before and after editing with white and calculate SSIM Wang et al. [2004] to evaluate this dimension.

### 3.1.2 Low-level Editing

Unlike high-level editing, low-level editing instructions are simpler, lacking specifications regarding object size, orientation, or color. Various low-level editing tasks Wang et al. [2023a], Chen et al. [2023a], Sanghvi et al. [2023], Chen et al. [2023b], Wu et al. [2023c], Guo et al. [2023a], Kong et al. [2022] have undergone extensive development over the years, resulting in a relatively mature evaluation system. Therefore, for low-level editing, we employ the widely recognized metric, namely SSIM Wang et al. [2004], to evaluate the editing quality.

**Deblurring.** Deblurring encompasses the procedure of mitigating or eliminating blur from images, resulting in enhanced clarity and sharpness.

**Haze Removal.** Haze removal entails the elimination or reduction of atmospheric haze or fog from images, augmenting visibility and reinstating the true colors and intricate details of the scene.

**Lowlight Enhancement.** Lowlight enhancement refers to the process of improving the quality of images captured in low-light conditions, enhancing brightness, and reducing noise.

**Noise Removal.** Noise removal involves the reduction or elimination of unwanted noises in images, resulting in cleaner and more visually appealing visuals.

**Rain Removal.** Rain removal aims to eliminate or reduce the visual effects of raindrops or rain streaks from images, improving clarity and restoring the original appearance.

**Shadow Removal.** Shadow removal refers to reducing or eliminating unwanted shadows from images, enhancing visibility, and improving overall image quality.

**Snow Removal.** The goal of Snow Removal is to effectively reduce or eliminate snow from images.

**Watermark Removal.** Watermark removal involves the removal or elimination of embedded watermarks from images, restoring the original appearance without the presence of the watermark.

### 3.2 Human Annotation

**Data Annotation.** We meticulously curated approximately 140 images from publicly available datasets Lin et al. [2014], Guo et al. [2023b], Martin et al. [2001], Chen et al. [2021], Ancuti et al. [2019], Liu et al. [2021b,a], Qu et al. [2017], Nah et al. [2017], Shen et al. [2019], Wei et al. [2018] for each evaluation dimension of I²EBench. The distribution of the image count for each dimension is illustrated in Fig. 3 (c). These images were then meticulously annotated with textual editing instructions by human annotators, namely original instructions. However, instructions provided by

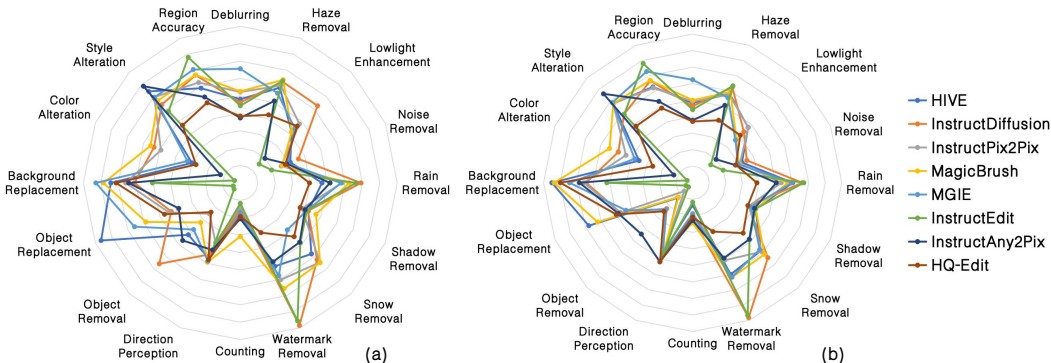

Figure 4: Comparison of radar charts for I²EBench scores in different dimensions using (a) original instructions and (b) diverse instructions.

human annotators usually followed a singular sentence pattern. For instance, the prevalent sentence pattern for the object removal dimension was typically "remove ⋯ from the image". To foster increased diversity, we employed ChatGPT Achiam et al. [2023] to effectively rewrite the original instructions. Fig. 3 (a) and (b) present the word cloud visualizations of the original and diverse instructions, respectively. Additionally, we also annotate a category for each instruction, such as animal, object, scenery, plant, human, and global.

**Evaluation Annotation**. The evaluation process for I²EBench encompasses two distinct categories. The first category employs conventional metrics to assess various dimensions. For style alteration dimension, we utilize the CLIP score as a standard metric, which doesn't need any additional evaluation annotations. In the second category, we utilize GPT-4V to evaluate the quality of editing. To facilitate this evaluation, we enlisted the expertise of human annotators to annotate questions specifically designed for GPT-4V, along with corresponding standard answers. For instance, let's consider the counting dimension and the instruction "Add a cat to the shoe rack". In this particular case, the annotated question provided by the human annotators is "How many cats are there on the shoe rack?", and the corresponding annotated answer is "One".

### 3.3 Human Evaluation

The primary objective of the human evaluation is to ascertain the correlation between human perception and the I²EBench score. To achieve this, we present human evaluators with a textual instruction $T$, an input image $V_I$, and a set of edited images $\{V_1, V_2, \cdots, V_M\}$ generated by $M$ different IIE models. The evaluators are then tasked with ranking the results based on their judgment. More specifically, we sample $N$ images for each evaluation dimension, leading to a comprehensive collection of $N \times 16 \times 2$ edited image comparisons. Within each comparison, evaluators are presented with $M$ edited images to assess and rank in relation to one another. We assign a human score to each model based on its ranking among the $M$ models. Specifically, the model ranked first among the $M$ models receives a human score of $M$, while the model ranked last among the $M$ models receives a human score of 1. Additionally, the model ranked $k$ among the $M$ models is assigned a human score of $M - k + 1$. To determine the human score for each dimension, we calculate the average of the human scores across all samples within that dimension. Thus, the human score for each model ranges from 1 to $M$.

## 4 Experiments

**Dimension Evaluation.** For each image and instruction, we utilize official codes from various models for image editing. We calculate the I²EBench scores following the methodology described in Sec.3.1. The I²EBench scores for original and diverse instructions are presented in Fig. 4, Tab.1, and Tab. 2, respectively. Our observations reveal that no single model achieves the best performance across all evaluation dimensions. Regarding low-level editing, InstructDiffusion Geng et al. [2023] demonstrates superior results. It attains the highest scores in 4 out of 7 low-level editing evaluation dimensions when using original instructions, and 3 out of 7 when using diverse instructions. For

Table 1: I²EBench evaluation results per dimension using original instructions. *Exp Min* and *Exp Max* denote the minimum and maximum values of all samples for each evaluation dimension.

| | | | | Low-level Editing | | | | |
|---|---|---|---|---|---|---|---|---|
| Model | Deblurring | Haze Removal | Lowlight Enhancement | Noise Removal | Rain Removal | Shadow Removal | Snow Removal | Watermark Removal |
| HIVE Zhang et al. [2023a] | 44.25 | 54.89 | 37.61 | 24.59 | 45.47 | 37.61 | 51.49 | 49.99 |
| InstructDiffusion Geng et al. [2023] | 42.48 | 58.45 | **56.61** | **28.60** | **67.20** | 37.43 | 55.65 | **85.49** |
| InstructPix2Pix Brooks et al. [2023] | 48.03 | 56.15 | 43.32 | 20.11 | 56.64 | 34.19 | 57.59 | 58.12 |
| MagicBrush Zhang et al. [2024a] | 48.38 | **59.46** | 37.71 | 20.59 | 60.60 | **41.91** | **57.81** | 63.33 |
| MGIE Fu et al. [2024] | **60.30** | 51.75 | 39.99 | 23.25 | 56.00 | 36.91 | 34.04 | 55.53 |
| InstructEdit Wang et al. [2023b] | 40.77 | 58.85 | 13.83 | 15.40 | 64.44 | 36.88 | 43.45 | 82.68 |
| InstructAny2Pix Li et al. [2023c] | 34.34 | 47.27 | 18.03 | 22.89 | 49.94 | 35.84 | 42.97 | 47.28 |
| HQ-Edit Hui et al. [2024] | 35.27 | 39.25 | 41.71 | 22.13 | 38.52 | 33.13 | 38.97 | 29.80 |
| Exp Min | 13.79 | 12.66 | 0.09 | 0.79 | 7.38 | 1.05 | 2.18 | 1.34 |
| Exp Max | 91.94 | 92.70 | 89.60 | 77.00 | 96.11 | 89.19 | 89.26 | 96.42 |

| | | | | High-level Editing | | | | |
|---|---|---|---|---|---|---|---|---|
| Model | Counting | Direction Perception | Object Removal | Object Replacement | Background Replacement | Color Alteration | Style Alteration | Region Accuracy |
| HIVE Zhang et al. [2023a] | 18.57 | 47.14 | 42.14 | **86.43** | 74.29 | 30.00 | 25.32 | 58.15 |
| InstructDiffusion Geng et al. [2023] | 15.00 | 44.29 | **65.71** | 42.86 | 60.71 | 53.57 | 21.69 | 66.18 |
| InstructPix2Pix Brooks et al. [2023] | 13.57 | 37.14 | 25.00 | 44.29 | 65.71 | 49.29 | 23.76 | 61.63 |
| MagicBrush Zhang et al. [2024a] | **30.71** | **49.29** | 32.14 | 58.57 | 78.57 | **55.71** | 22.78 | 66.34 |
| MGIE Fu et al. [2024] | 17.14 | 48.57 | 37.86 | 65.71 | **82.86** | 32.86 | 23.68 | 69.60 |
| InstructEdit Wang et al. [2023b] | 11.76 | 41.73 | 5.04 | 4.41 | 50.36 | 3.62 | 19.83 | **77.08** |
| InstructAny2Pix Li et al. [2023c] | 20.59 | 41.73 | 46.76 | 38.24 | 64.03 | 12.32 | **26.76** | 52.75 |
| HQ-Edit Hui et al. [2024] | 19.26 | 47.79 | 23.74 | 47.06 | 71.22 | 27.54 | 15.96 | 49.21 |
| Exp Min | 0.00 | 0.00 | 0.00 | 0.00 | 0.00 | 0.00 | 12.96 | 6.41 |
| Exp Max | 100.00 | 100.00 | 100.00 | 100.00 | 100.00 | 100.00 | 33.84 | 98.70 |

Table 2: I²EBench evaluation results per dimension using diverse instructions.

| | | | | Low-level Editing | | | | |
|---|---|---|---|---|---|---|---|---|
| Model | Deblurring | Haze Removal | Lowlight Enhancement | Noise Removal | Rain Removal | Shadow Removal | Snow Removal | Watermark Removal |
| HIVE Zhang et al. [2023a] | 44.41 | 54.09 | 42.78 | 25.51 | 58.59 | 36.69 | 51.92 | 57.88 |
| InstructDiffusion Geng et al. [2023] | 42.62 | 58.01 | 39.47 | **28.06** | 64.18 | 32.54 | **57.30** | **85.14** |
| InstructPix2Pix Brooks et al. [2023] | 45.24 | 53.52 | **42.88** | 24.49 | 51.86 | 32.79 | 52.67 | 48.91 |
| MagicBrush Zhang et al. [2024a] | 45.96 | 55.11 | 33.74 | 23.91 | 55.77 | **36.73** | 54.68 | 59.76 |
| MGIE Fu et al. [2024] | **57.33** | 51.61 | 32.96 | 23.49 | 58.27 | 34.07 | 51.02 | 59.64 |
| InstructEdit Wang et al. [2023b] | 40.66 | **58.89** | 13.92 | 15.81 | **65.08** | 36.66 | 43.34 | 83.68 |
| InstructAny2Pix Li et al. [2023c] | 34.77 | 47.00 | 18.09 | 22.18 | 48.92 | 36.04 | 43.13 | 47.58 |
| HQ-Edit Hui et al. [2024] | 34.11 | 37.95 | 36.76 | 22.38 | 37.60 | 32.17 | 38.45 | 30.83 |
| Exp Min | 6.32 | 3.67 | 0.60 | 0.03 | 7.22 | 1.46 | 3.78 | 2.58 |
| Exp Max | 88.17 | 92.69 | 90.34 | 79.29 | 97.03 | 86.27 | 82.24 | 96.39 |

| | | | | High-level Editing | | | | |
|---|---|---|---|---|---|---|---|---|
| Model | Counting | Direction Perception | Object Removal | Object Replacement | Background Replacement | Color Alteration | Style Alteration | Region Accuracy |
| HIVE Zhang et al. [2023a] | 13.57 | 43.57 | 12.86 | **67.86** | **85.00** | 35.00 | 23.08 | 61.97 |
| InstructDiffusion Geng et al. [2023] | 21.43 | 47.86 | 22.14 | 47.14 | 64.29 | 48.57 | 19.96 | 65.92 |
| InstructPix2Pix Brooks et al. [2023] | 18.57 | 47.86 | 7.14 | 47.14 | 65.71 | 43.57 | 23.13 | 61.32 |
| MagicBrush Zhang et al. [2024a] | **24.29** | 45.71 | 12.14 | 62.14 | 83.57 | **54.29** | 23.08 | 66.21 |
| MGIE Fu et al. [2024] | 19.29 | 47.14 | 22.86 | 43.57 | 74.29 | 37.86 | 23.36 | 71.89 |
| InstructEdit Wang et al. [2023b] | 11.76 | 46.04 | 3.60 | 4.41 | 51.80 | 3.62 | 19.91 | **77.08** |
| InstructAny2Pix Li et al. [2023c] | 22.79 | **51.80** | **43.88** | 48.53 | 68.35 | 12.32 | **25.93** | 52.61 |
| HQ-Edit Hui et al. [2024] | 20.74 | 51.47 | 24.46 | 50.00 | 79.86 | 26.09 | 16.48 | 48.29 |
| Exp Min | 0.00 | 0.00 | 0.00 | 0.00 | 0.00 | 0.00 | 10.62 | 9.79 |
| Exp Max | 100.00 | 100.00 | 100.00 | 100.00 | 100.00 | 100.00 | 34.06 | 98.68 |

high-level editing, both MagicBrush Zhang et al. [2024a] and InstructAny2Pix Li et al. [2023c] perform impressively. MagicBrush achieves the highest scores in 3 evaluation dimensions using original instructions, while InstructAny2Pix achieves the highest scores in 3 dimensions using diverse instructions. In the deblurring dimensions, MGIE Fu et al. [2024] stands out significantly. It surpasses the second-place model by 11.92 when using original instructions and by 11.37 when using diverse instructions.

**Human Evaluation.** We ranked different models based on their I²EBench scores and computed I²EBench rank scores using the methodology described in Sec. 3.3. Given that both I²EBench rank scores and human scores range from 1 to 8, a direct comparison can be made between them. Therefore, we conducted correlation analyses and visually presented the results in Fig. 5. Significant positive correlations were observed between the I²EBench rank score and the human score across all

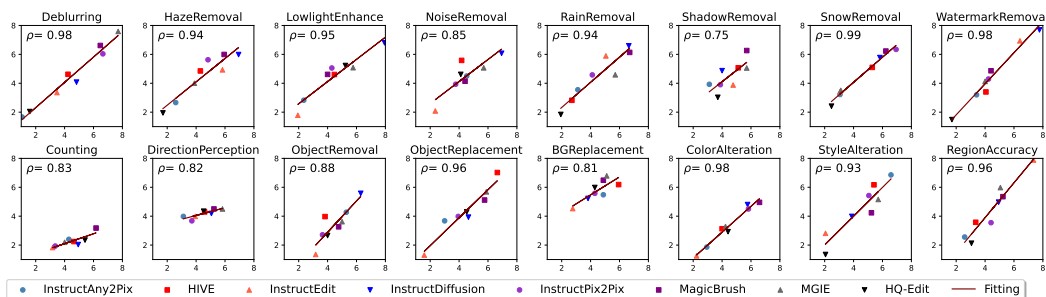

Figure 5: Alignment between I²EBench rank scores (Y-axis) and human scores (X-axis).

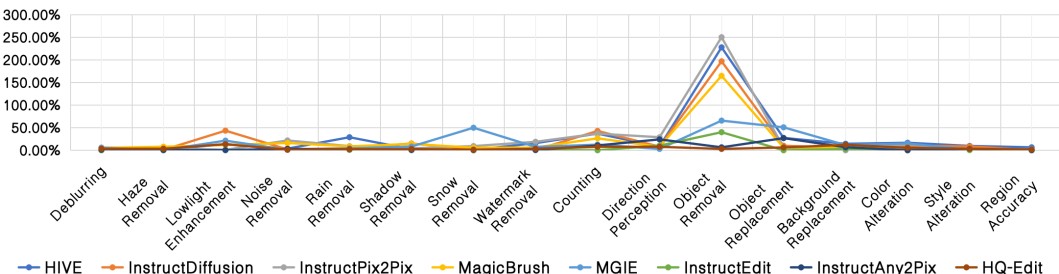

Figure 6: I²EBench change rate using original instructions and diverse instructions.

dimensions. These findings offer strong evidence supporting the alignment between our proposed benchmark and human perception.

## 5 Insights

**The editing ability across different dimensions is not robust:** Our observations indicate that no single model excels in all evaluation dimensions. This implies that different IIE models have varying strengths in terms of their editing abilities across different dimensions. Thus, it is crucial to acknowledge this limitation and focus on developing an IIE model that demonstrates consistent and competent performance across all dimensions. *Future research efforts should prioritize the creation of a robust and versatile IIE model that can effectively handle a wide range of editing tasks across diverse dimensions.*

**The editing ability of different instructions is not robust:** To evaluate the robustness of editing models when provided with different instructions, we propose a metric called I²EBench change rate. This metric is defined as follows:

$$S^i = \frac{|S_o^i - S_d^i|}{\text{MIN}(S_o^i, S_d^i)},\tag{1}$$

where $S_o^i$ and $S_d^i$ represent the I²EBench scores of the $i$-th evaluation dimension when using original and diverse instructions, respectively. The value of $S^i$ indicates the I²EBench change rate for the $i$-th evaluation dimension. As illustrated in Fig. 6, when it comes to the object removal dimension, InstructPix2PixBrooks et al. [2023], HIVE Zhang et al. [2023a], InstructionDiffusion Geng et al. [2023], and MagicBrush Zhang et al. [2024a] exhibit significant fluctuations in their performance using different instructions. On the other hand, the remaining models demonstrate relatively stable performance across different instructions. One notable distinction between these two categories of models is that the latter employs LLM Achiam et al. [2023], Touvron et al. [2023] or MLLM Liu et al. [2023b,a] to comprehend instructions, which enhances their resilience to variations in instructions. *Given the unpredictable and diverse nature of user editing instructions, it is crucial to develop an editing model that can effectively handle instructions with varying levels of complexity.*

**The editing ability for different categories is not robust:** As illustrated in Fig. 7, we have observed distinct variations in the performance of different categories. Notably, the "Scenery" and "Global"

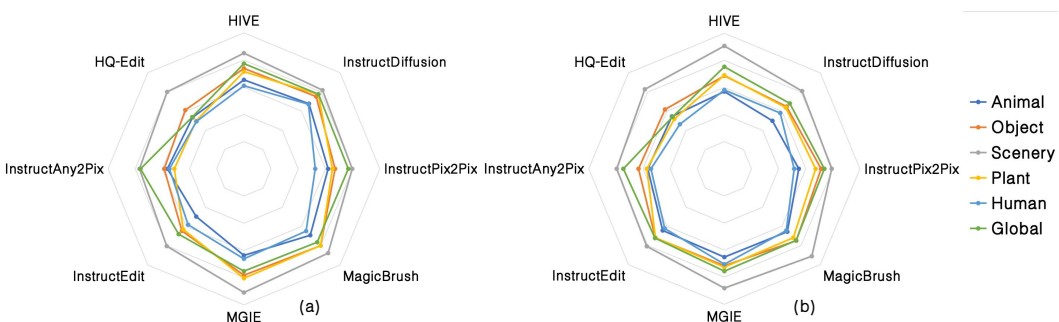

Figure 7: Comparison of radar charts for I$^2$EBench scores in different categories using (a) original instructions and (b) diverse instructions. The scores of all dimensions are normalized and averaged.

categories consistently demonstrate superior performance compared to the other categories across all the IIE models we evaluated. This discrepancy can be attributed to the inherent inclination of the "Scenery" and "Global" categories towards global editing, which diminishes the necessity for precise target object localization. *Given these findings, it is crucial to prioritize the simultaneous consideration of various editing content in future research endeavors.*

## 6    Conclusions

In this paper, we present I$^2$EBench, a comprehensive benchmark specifically designed for instruction-based image editing (IIE). Our benchmark includes a substantial dataset of over 2000+ images and more than 4000+ instructions, covering 16 distinct evaluation dimensions. To evaluate the effectiveness of I$^2$EBench, we conduct experiments using 8 open-source IIE models. Additionally, we complement these experiments with meticulous human evaluations to establish the correlation between I$^2$EBench scores and human perception. Based on the observations derived from I$^2$EBench, we provide valuable insights and recommendations for advancing IIE models. We hope the proposed I$^2$EBench to serve as an indispensable asset, playing a pivotal role in fostering the advancement of IIE models and assessing their efficacy.

## Acknowledge

This work was supported by National Key R&D Program of China (No.2023YFB4502804), the National Science Fund for Distinguished Young Scholars (No.62025603), the National Natural Science Foundation of China (No. U22B2051, No. U21B2037, No. 62072389, No. 62302411), the Natural Science Foundation of Fujian Province of China (No.2021J06003), and China Postdoctoral Science Foundation (No. 2023M732948).

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
