# OpenReview forum: "I2EBench: A Comprehensive Benchmark for Instruction-based Image Editing"
_NeurIPS.cc/2024/Conference — NeurIPS 2024 poster_

### Official Review · Reviewer_HHca · 2024-06-28

**Soundness:** 2
**Presentation:** 3
**Contribution:** 2
**Rating:** 6
**Confidence:** 5

**Summary:**

This paper presents a comprehensive benchmark for instructional image editing. The benchmark contains a high-quality dataset with over 2000 images and 4000 instructions. In addition, the benchmark presents a new evaluation pipeline that leverages GPT to act as the judge to validate the performance of different instructional image editing methods. Within the evaluation, there are two major factor groups considering both high-level and low-level editing.

**Strengths:**

Due to the lack of high-quality benchmarks in image editing, this work fills the gap and is of great significance to the community. The evaluation pipeline is reasonable and clear to follow. The idea of using multimodal large language models to evaluate image editing results is interesting. The overall presentation is clear.

**Weaknesses:**

1. The provided Google Drive link cannot be opened, making the benchmark images inaccessible for review.

2. Image editing evaluation remains a challenge due to few benchmarks. However, the authors ignore comparing their work with a related work [1] due to similar technical pipelines used with that work. Both this and that work first perform human collection, then automated evaluation using GPT, then human evaluation, and then alignment evaluation. The authors should explain the missing comparison.

3. In the high-level editing, the authors did not discuss the evaluation dimension of action change or shape and size change, which are also quite essential editing types.

[1] Diffusion Model-Based Image Editing: A Survey. https://arxiv.org/abs/2402.17525

**Questions:**

I am confused about the definition of low-level image editing, which is actually restoration and enhancement.

First, low-level restoration tasks usually involve fine-grained operations with rich details. Can GPT really see the slight difference when two similar restored results only vary slightly in terms of visual observation or PSNR evaluation?

Second, many editing methods are not designed for low-level vision tasks. It is not appropriate to use them to perform these tasks. The authors should check more methods specially designed for these tasks.

I hope the authors can perform additional experiments to address the above concerns. For example, for low-light image enhancement, using two leading methods (such as [2] and [3]) to enhance one image with similar outputs visually (or similar PSNR) and then using GPT for evaluation.

[2] Retinexformer: One-stage Retinex-based Transformer for Low-light Image Enhancement (https://arxiv.org/abs/2303.06705)

[3] Low-Light Image Enhancement with Wavelet-based Diffusion Models (https://arxiv.org/abs/2306.00306)

**Limitations:**

The paper should be submitted to the NeurIPS Dataset and Benchmark track.

---

> ### Author Rebuttal · Authors · 2024-08-04
>
> We would like to extend our heartfelt gratitude for your thoughtful and encouraging feedback on our paper. We are deeply appreciative of your commendation, recognizing our work as a significant contribution that fills a notable gap in high-quality benchmarks for image editing. Your praise regarding the reasonableness and clarity of the evaluation pipeline, as well as the innovative use of multimodal large language models to assess image editing results, is highly encouraging. Additionally, we are grateful for your positive remarks on the overall clarity of our presentation.
>
> Now, let us address the specific weaknesses and questions you have highlighted and provide further clarification:
>
> > ***Q1:** The provided Google Drive link cannot be opened, making the benchmark images inaccessible for review.*
> >
>
> **A1:**
>
> We sincerely apologize for the inconvenience caused by the inaccessibility of the Google Drive link. Upon investigation, we discovered that our anonymous Google account was mistakenly suspended by Google's system on June 21, 2024, resulting in the data on Google Drive becoming inaccessible. We have since appealed this suspension, and the Google Drive link is now accessible once again. If you have any further questions or encounter any other issues, please do not hesitate to contact us.
>
> > Q2: Image editing evaluation remains a challenge due to few benchmarks. However, the authors ignore comparing their work with a related work [1] due to similar technical pipelines used with that work.
> >
> >
> > *[1] Diffusion Model-Based Image Editing: A Survey.*
> >
>
> **A2:**
>
> Thank you for pointing out the relevant and intriguing paper. We apologize for not referencing and comparing our work with this study prior to submission. We will incorporate a thorough discussion and comparison with this work in our revised manuscript. Specifically, the comparison between our benchmark and the one presented in [1] can be outlined as follows:
>
> 1. **Evaluation Methodology:**
>     - **Our Approach:** For each evaluation dimension, we meticulously designed questions to compare GPT’s responses with annotated answers to assess the accuracy of the edits.
>     - **[1]’s Approach:** The related work uses GPT to score the editing results.
> 2. **Number of Dimensions:**
>     - **Our Benchmark:** We propose a comprehensive benchmark with 16 high-level and low-level dimensions to thoroughly evaluate different types of edits.
>     - **[1]’s Benchmark:** The related work covers 8 dimensions.
> 3. **Number of Images per Dimension:**
>     - **Our Benchmark:** I2EBench includes approximately 120 images for each dimension, providing robust validation for each type of edit.
>     - **[1]’s Benchmark:** The related work includes 50 images per dimension.
>
> We will ensure these detailed discussions are included in the final version of our manuscript. Your suggestion will undoubtedly contribute to a more comprehensive and complete presentation of our work.
>
> > ***Q3:** In the high-level editing, the authors did not discuss the evaluation dimension of action change or shape and size change, which are also quite essential editing types.*
> >
>
> **A3:**
>
> Thank you for your valuable feedback. You are absolutely correct that the evaluation dimensions of action change and shape and size change are crucial for a comprehensive assessment of high-level editing. We appreciate your insightful recommendation.
>
> In light of your suggestion and the mentioned related works[1], we will actively incorporate these evaluation dimensions into our benchmark in future iterations. This will ensure that all essential aspects of high-level editing are thoroughly evaluated, ultimately contributing to more accurate and useful benchmarking for the research community.
>
> > ***Q4:** First, low-level restoration tasks usually involve fine-grained operations with rich details. Can GPT really see the slight difference when two similar restored results only vary slightly in terms of visual observation or PSNR evaluation? …  For example, for low-light image enhancement, using two leading methodsto enhance one image with similar outputs visually (or similar PSNR) and then using GPT for evaluation.*
> >
>
> **A4:**
>
> Thank you for your insightful question. It appears there may have been some misunderstanding regarding our evaluation methodology for low-level restoration tasks. As indicated in lines 186-187 of our manuscript, we employed the Structural Similarity Index (SSIM) between the ground truth image and the edited image to measure the quality of low-level edits. Consequently, we did not rely on GPT-4V for this aspect of the evaluation. Therefore, whether GPT can discern slight differences when two similar restored results vary minimally in visual observation or PSNR evaluation does not impact the accuracy of our low-level evaluation.
>
> To prevent further misunderstandings, we will clarify our evaluation methodology as detailed on lines 186-187 in the revised manuscript.
>
> > ***Q5:** Second, many editing methods are not designed for low-level vision tasks. It is not appropriate to use them to perform these tasks.*
> >
>
> **A5:**
>
> Thank you for your insightful suggestion.
>
> For various independent low-level tasks, there are already well-established datasets in existence. Models specifically designed for these tasks can be appropriately evaluated using these datasets.
>
> However, the advent of large models capable of handling multiple tasks has become a significant trend in the field. Our primary objective is to encourage research on versatile models that can support both high-quality high-level and low-level image editing. Therefore, in designing our benchmark, we focused on assessing the capabilities of general-purpose editing models across a range of evaluation dimensions, encompassing both high-level and low-level editing capabilities.

---

> > ### Comment · Reviewer_HHca · 2024-08-10
> >
> > Thanks for the response. Most of my concerns are addressed. Hence I increase my rating.

---

> > > ### Author Response · Authors · 2024-08-11
> > > **Response to Reviewer HHca**
> > >
> > > We are delighted that our response has addressed your concerns. Thank you for increasing your rating.

---

> ### Author Response · Authors · 2024-08-09
> **Sincere Request for Further Discussions**
>
> Dear Reviewer HHca,
>
> Thank you for your invaluable efforts and constructive feedback on our manuscript.
>
> As the discussion period draws to a close, we eagerly anticipate your thoughts on our response. We sincerely hope that our response meets your expectations. If there are any remaining concerns or aspects that require clarification, we are ready to address them as soon as possible.
>
> Best regards,
>
> The Authors

---

### Official Review · Reviewer_A2Kh · 2024-06-30

**Soundness:** 4
**Presentation:** 4
**Contribution:** 4
**Rating:** 8
**Confidence:** 5

**Summary:**

This paper proposes I2EBench, which is a new benchmark for evaluating Instruction-based Image Editing (IIE) models. It offers a large dataset with over 2,000 images and 4,000 instructions across 16 detailed evaluation dimensions. The benchmark is designed to assess image editing quality automatically and aligns with human perception through extensive user studies. I2EBench aims to provide insights for improving IIE models and will be open-sourced for community use.

**Strengths:**

I commend the authors for their insightful paper, particularly the proposed benchmark, which is remarkably comprehensive and has the potential to significantly advance the field of instruction-based image editing (IIE).
1.The benchmark addresses a wide array of evaluation dimensions, offering a holistic and multifaceted assessment of IIE model capabilities. This comprehensive approach ensures that the strengths and weaknesses of various models are thoroughly examined from multiple perspectives.
2.It strongly emphasizes aligning with human perception, ensuring the benchmark's relevance to human preference.
3.The large collection of images and instructions forms a solid foundation for comprehensive testing. This large and diverse dataset provides a robust foundation for thorough and rigorous testing, enabling models to be evaluated across a broad spectrum of scenarios.

**Weaknesses:**

Although the paper proposes a amazing benchmark, I think the following points can further improve the paper:
1.The results of multiple models on the benchmark were presented in the paper; however, the specifics of their evaluation were not described in sufficient detail. For instance, information regarding the hyperparameters used for the evaluation process and the types of GPUs employed for model testing would provide greater clarity and reproducibility.
2.The rationale behind the use of GPT-4V for supplementary evaluation requires further elucidation. Additionally, whether alternative models could serve as suitable substitutes for GPT-4V?
3.In the section on Instruction, the contributions of the paper are not sufficiently highlighted. A more thorough summarization of the paper's contributions would significantly aid in conveying its impact to the readers.

**Questions:**

I would like to inquire if the I2EBench tool or framework will be open-sourced in the future. This information is crucial for understanding its potential accessibility and contributions to the community.

**Limitations:**

Yes. The limitations of the paper described by the author in the appendix.

---

> ### Author Rebuttal · Authors · 2024-08-04
>
> We would like to extend our heartfelt gratitude for your thoughtful and encouraging feedback on our paper. We are deeply appreciative of your commendation, recognizing our work as insightful and valuable. Thank you for acknowledging that our benchmark encompasses a wide range of evaluation dimensions and places a strong emphasis on alignment with human perception. Additionally, we are grateful for your praise regarding the extensive collection of images and instructions, which indeed forms a robust foundation for comprehensive testing.
>
> Now, let us address the specific weaknesses and questions you have highlighted and provide further clarification:
>
> > ***Q1:** The results of multiple models on the benchmark were presented in the paper; however, the specifics of their evaluation were not described in sufficient detail. For instance, information regarding the hyperparameters used for the evaluation process and the types of GPUs employed for model testing would provide greater clarity and reproducibility.*
> >
>
> **A1:**
>
> Thank you for your valuable feedback. We utilized the hyperparameter settings and official weights as specified in the original papers for each model. All editing results were generated using A800 GPUs.
>
> We will include these implementation details in the revised version of our paper to enhance transparency and facilitate the reproducibility of our results.
>
> > ***Q2:** The rationale behind the use of GPT-4V for supplementary evaluation requires further elucidation. Additionally, whether alternative models could serve as suitable substitutes for GPT-4V?*
> >
>
> **A2:**
>
> Thank you for your insightful question.
>
> Our decision to use GPT-4V in our supplementary evaluation was based on its status as the most advanced model available at the time of our study. At that point, GPT-4o had not yet been proposed. We chose GPT-4V for its sophisticated capabilities, including its nuanced understanding of language and proficiency in handling complex tasks. As technology continues to evolve, it is indeed possible that other multimodal large language models may emerge, offering even more accurate assessments in the future.
>
> > ***Q3:** In the section on Instruction, the contributions of the paper are not sufficiently highlighted. A more thorough summarization of the paper's contributions would significantly aid in conveying its impact to the readers.*
> >
>
> **A3:**
>
> Thank you for your valuable feedback. In light of your comments, we have outlined our key contributions as follows:
>
> - We have proposed a comprehensive benchmark encompassing 16 evaluation dimensions specifically designed to assess instruction-based image editing (IIE) tasks.
> - We have conducted extensive experiments on eight popular IIE models, accompanied by a thorough analysis of their performance.
> - We have implemented Alignment Verification, demonstrating that our benchmark scores are aligned with human perception.
>
> We will incorporate this summarized list of contributions into the revised manuscript to ensure that the readers can clearly understand the impact and significance of our work.
>
> > ***Q4:** I would like to inquire if the I2EBench tool or framework will be open-sourced in the future. This information is crucial for understanding its potential accessibility and contributions to the community.*
> >
>
> **A4:**
>
> Thank you for your suggestion. The evaluation scripts for I2EBench have already been included in the supplementary materials. We are committed to open-sourcing all I2EBench codes and datasets to foster progress in this field. This initiative will ensure the accessibility and usability of our tool for the broader research community, promoting further advancements and collaborative developments.

---

> ### Author Response · Authors · 2024-08-09
> **Sincere Request for Further Discussions**
>
> Dear Reviewer A2Kh,
>
> Thank you for your invaluable efforts and constructive feedback on our manuscript.
>
> As the discussion period draws to a close, we eagerly anticipate your thoughts on our response. We sincerely hope that our response meets your expectations. If there are any remaining concerns or aspects that require clarification, we are ready to address them as soon as possible.
>
> Best regards,
>
> The Authors

---

> > ### Comment · Area_Chair_71VD · 2024-08-13
> >
> > Dear Reviewer A2Kh:
> >
> > Thanks for reviewing this work. Would you mind to check authors' feedback and see if it resolves your concerns or you may have further comments?
> >
> > Best, AC

---

> ### Comment · Reviewer_A2Kh · 2024-08-14
>
> Thank you for the author's response. All my concerns have been addressed. After carefully reviewing all the reviewers' comments and the author's response, I found that all reviewers acknowledge the value of I2EBench. I further believe that I2EBench can significantly contribute to the image editing community. Therefore, I will further raise my score and hope the authors can incorporate all the feedback into the revised version.

---

> > ### Author Response · Authors · 2024-08-14
> > **Response to Reviewer A2Kh**
> >
> > We greatly appreciate your efforts in reviewing and your recognition of I2EBench's contribution to the community. We are pleased that our response has resolved your concerns.

---

### Official Review · Reviewer_7DEf · 2024-07-01

**Soundness:** 4
**Presentation:** 4
**Contribution:** 4
**Rating:** 8
**Confidence:** 5

**Summary:**

The paper addresses the challenge of evaluating models in the field of Instruction-based Image Editing (IIE) by proposing a comprehensive benchmark called I2EBench. It features: 1) Comprehensive Evaluation: Covers 16 evaluation dimensions for a thorough assessment of IIE models. 2) Human Perception Alignment: Includes extensive user studies to ensure relevance to human preferences. 3) Research Insights: Provides analysis of strengths and weaknesses of existing IIE models to guide future development. I2EBench will be open-sourced, including all instructions, images, annotations, and a script for evaluating new models.

**Strengths:**

1 The supplementary materials provided by the authors offer an in-depth explanation of the evaluation process, which is very beneficial for readers seeking to understand the evaluation details of I2EBench thoroughly.

2 The benchmark proposed in the paper covers a wide range of evaluation dimensions, providing a holistic and multi-faceted assessment of IIE model capabilities.

3 The authors commit to open-sourcing I2EBench, including all relevant resources. This openness will facilitate fair comparisons and knowledge sharing within the community.

4 By systematically evaluating the models, the paper provides valuable research insights that can guide future model architecture design and data selection strategies.

5 The inclusion of user studies in the evaluation process adds depth, ensuring that the evaluation results accurately reflect the real-world experiences of end-users.

6 The benchmark encompasses multiple types of image editing tasks, including both high-level and low-level editing, which makes it versatile and comprehensive.

7 A large number of images and instructions are provided, forming a solid foundation for comprehensive testing of the models.

**Weaknesses:**

1 If the research code or data is not made publicly available, it may limit the usability of the benchmark for other researchers.

2 The paper only provides radar charts for the category experiments. Including the corresponding quantitative data would make the comparisons more precise and understandable.

3 I observed a significant performance gap in the Object Removal dimension between using the original instruction and diverse instruction. The authors could use methods like CLIP similarity or Jaccard similarity to measure the similarity between the original and diverse instructions. This could help determine whether the performance variance is due to significant changes in the instructions or due to the sensitivity of some models to specific vocabulary changes in the Object Removal instructions.

4 The font size in Figure 3 (c) is too small, which might hinder readers from clearly viewing the information presented in the figure.

**Questions:**

1 Why did the authors choose to sample high-level editing images from the COCO dataset, while most low-level editing images are sampled from existing low-level datasets?

Others please ref to the weaknesses.

**Limitations:**

yes

---

> ### Author Rebuttal · Authors · 2024-08-04
>
> We would like to extend our heartfelt gratitude for your thoughtful feedback on our paper. We sincerely appreciate your recognition of its strengths, including the comprehensive explanation of the evaluation process, the wide array of evaluation dimensions, the provision of evaluation code, and the valuable research insights we have provided. Additionally, we are thankful for your acknowledgment that I2EBench is a holistic benchmark encompassing both low-level and high-level editing tasks.
>
> Now, let us address the specific weaknesses and questions you have highlighted, and provide further clarification:
>
> > ***Q1:** If the research code or data is not made publicly available, it may limit the usability of the benchmark for other researchers.*
> >
>
> **A1:**
>
> Thank you for raising this important concern.
>
> We fully appreciate that the accessibility of research code and data is crucial for the usability and reproducibility of our benchmark by other researchers. In response to this, we have included the code in the supplementary materials accompanying our submission. Additionally, we are committed to fully open-sourcing both the code and data.
>
> > ***Q2:** The paper only provides radar charts for the category experiments. Including the corresponding quantitative data would make the comparisons more precise and understandable.*
> >
>
> **A2:**
>
> Thank you for your insightful suggestion.
>
> To provide a more precise and comprehensible comparison of our benchmark's performance across different categories, we have included the corresponding quantitative data. **Tab. III and Tab. IV of the rebuttal pdf** present the quantitative results for different model categories, supplementing the radar charts displayed in Figure 7. These detailed metrics will be incorporated into the supplementary materials of the final version of our paper to enhance clarity and further substantiate our findings.
>
> > ***Q3:** I observed a significant performance gap in the Object Removal dimension between using the original instruction and diverse instruction. The authors could use methods like CLIP similarity or Jaccard similarity to measure the similarity between the original and diverse instructions. This could help determine whether the performance variance is due to significant changes in the instructions or due to the sensitivity of some models to specific vocabulary changes in the Object Removal instructions.*
> >
>
> **A3:**
>
> Thank you for your valuable suggestion.
>
> Following your recommendation, we have employed multiple metrics to measure the similarity between the original and diverse instructions, including CLIP cosine similarity, Jaccard similarity, TF-IDF, Word2Vec, and FastText. **Tab. V of the rebuttal pdf** presents the similarity scores, illustrating that there are no significant abnormalities across these metrics when comparing the original and diverse instructions in the Object Removal dimension. Thus, the observed performance gap in the Object Removal dimension is not due to textual discrepancies. Instead, it indicates that these models exhibit a lack of robustness in interpreting the instructions for this particular task.
>
> > ***Q4:** The font size in Figure 3 (c) is too small, which might hinder readers from clearly viewing the information presented in the figure.*
> >
>
> **A4:**
>
> Thank you for your valuable feedback.
>
> We understand that the readability of figures is crucial for conveying information effectively. In response to your observation, we will revise Figure 3(c) to increase the font size, ensuring that all text is clearly legible.
>
> > **Q5:** Why did the authors choose to sample high-level editing images from the COCO dataset, while most low-level editing images are sampled from existing low-level datasets?
> >
>
> **A5:**
>
> Thank you for your insightful question.
>
> For low-level editing tasks, we sourced images from existing low-level datasets because these datasets provide ground truth (GT) images. The availability of GT images allows for a more precise evaluation by directly calculating metrics such as SSIM (Structural Similarity Index) between the GT images and the edited images, ensuring accuracy in assessment.
>
> In contrast, high-level editing tasks often lack clearly defined ground truth images, making it challenging to perform precise evaluations. Therefore, we chose to sample high-level editing images from the COCO dataset, a widely recognized and accepted dataset in the research community. The use of COCO ensures a broad and diverse range of images, facilitating a more representative assessment of high-level editing capabilities.

---

> > ### Comment · Reviewer_7DEf · 2024-08-08
> > **Response to Authors**
> >
> > Thank you for responding to my concerns.
> >
> > For A1, thanks for your commitment. I look forward to the development of I2EBench in the IIE field.
> >
> > For A2, I think quantitative tables can better show the absolute difference in performance than qualitative charts. I suggest replacing Figure 7 with a table.
> >
> > For A3, the response has resolved my issue regarding the performance gap in the object removal dimension. The response is reasonable and supported by experimental evidence.
> >
> > For A4 and A5, thanks for your response.
> >
> > ---
> >
> > This paper proposes a comprehensive benchmark to fill the gap in evaluating high-level and low-level image editing. Based on the author's response, which addressed my concerns, I have decided to increase my score from 6 to 8.

---

> > > ### Author Response · Authors · 2024-08-09
> > > **Response to Reviewer 7DEf**
> > >
> > > Thank you for acknowledging our work. We will open-source I2EBench in the near future and incorporate your suggestions to further refine our paper.

---

### Official Review · Reviewer_riYa · 2024-07-07

**Soundness:** 3
**Presentation:** 2
**Contribution:** 3
**Rating:** 5
**Confidence:** 5

**Summary:**

This paper proposes I2EBench, a comprehensive benchmark designed to automatically evaluate the quality of edited images produced by IIE models from multiple dimensions. I2EBench comprises 16 evaluation dimensions, covering both high-level and low-level aspects. Additionally, through user studies, the authors assess the alignment between the proposed benchmark and human perception. The I2EBench dataset consists of over 2000 images for editing, along with corresponding original images and diverse instructions.

**Strengths:**

1. The proposed I2EBench represents a significant advancement over previous works, providing a large and comprehensive benchmark for instruction-based image editing. This contribution will greatly benefit the research community.
2. When establishing I2EBench, the authors have thoughtfully included often overlooked low-level edits, such as rain removal, and have implemented thorough evaluations.
3. The paper is clearly written and easy to follow.

**Weaknesses:**

1. The technical novelty of the evaluation method is insufficient. A significant part of I2EBench's evaluation of high-level edits relies on GPT-4V, such as Direction Perception (line 158) and Object Removal (line 164). Therefore, the authors' contribution appears more like a new prompt engineering method.

2. The insights provided in Section 5 are not informative for the research community. The authors state that "the editing ability across different dimensions is not robust," and Figure 7 shows that current IIE methods perform better on high-level editing tasks (e.g., Object Removal) but struggle with low-level tasks (e.g., Shadow Removal). However, most existing editing datasets are high-level, making it difficult for IIE methods to learn low-level tasks during training. Therefore, the insights in Section 5 seems not constructive.

**Questions:**

1. Does I2EBench consider the aesthetic quality of image edits? For example, in Object Replacement (line 164), how does the benchmark evaluate whether the edited object is appropriately and naturally integrated into the image, rather than simply copied and pasted?

**Limitations:**

The authors adequately addressed the limitations in the manuscript.

---

> ### Author Rebuttal · Authors · 2024-08-04
>
> We would like to express our sincere gratitude for your positive feedback on our paper and for recognizing its strengths. We appreciate your agreement that I2EBench represents a significant advancement over previous works, greatly benefiting the research community. Additionally, we are thankful for your observation that we have thoughtfully included often overlooked low-level edits. We are also grateful for your praise regarding the clarity and readability of our paper.
>
> Now, let us address the specific weaknesses and questions you have raised and provide further clarification:
>
> > ***Q1:** The technical novelty of the evaluation method is insufficient.*
> >
>
> **A1:**
>
> Thank you very much for your insightful feedback. In fact, GPT-assisted evaluation methodologies are widely recognized and utilized in numerous influential published works [1,2,3], demonstrating considerable scientific validity. While it is true that we leverage GPT-4V for certain aspects of the evaluation, it is crucial to understand that our approach represents more than mere prompt engineering. To elucidate further, the effective utilization of GPT-4V in our benchmark necessitated several significant contributions:
>
> - **Image Selection:** For each evaluation dimension, we meticulously select and filter suitable images. For example, for the evaluation dimensions of "Object Removal" and "Object Replacement", it is necessary to ensure that there is at least one object in the image instead of a simple landscape photo. This rigorous curation process ensures the relevance and accuracy of the images used, thereby enhancing the integrity of our evaluation.
> - **Instruction Annotation:** Each image is annotated with a specific editing instruction that aligns precisely with the corresponding evaluation dimension. This careful annotation is essential for ensuring that the assessments are accurate and meaningful.
> - **Question-Answer Pairing for High-Level Edits:** For dimensions involving high-level edits, we annotate each image-instruction pair with corresponding question-answer pairs. The accuracy of the edits is then evaluated by comparing GPT-4V's responses to these annotated answers. This process allows us to rigorously assess the correctness and effectiveness of the edits.
>
> Thus, I2EBench should be viewed not merely as a prompt engineering method but as a well-designed, thoroughly annotated benchmark. It leverages GPT-4V to reduce evaluation costs and enable the automation of the evaluation process, which is in line with current practices in the field.
>
> *[1] Q-bench: A benchmark for general-purpose foundation models on low-level vision. ICLR. 2023.*
>
> *[2] Evalcrafter: Benchmarking and evaluating large video generation models. CVPR. 2024.*
>
> *[3] Judging llm-as-a-judge with mt-bench and chatbot arena. NeurIPS. 2023.*
>
> > ***Q2:** The insights provided in Section 5 are not informative for the research community.*
> >
>
> **A2:**
>
> Thank you for your constructive feedback.
>
> We agree that the current emphasis on high-level tasks in existing IIE datasets may limit the capacity of IIE methods to effectively learn and execute low-level editing tasks. This limitation actually reinforces the validity and reliability of our analysis. Our proposed I2EBench aims to address this gap by offering a comprehensive evaluation framework that includes both high-level and low-level editing tasks.
>
> Therefore, far from being non-informative, we believe our proposed insights serve as a crucial signal to the research community, highlighting the clear need for balanced dataset development and methodological advancements that can better handle low-level tasks. These insights provide a clear direction for future research and innovation, emphasizing areas that require further exploration and improvement. Your valuable comments have enabled us to more effectively convey the significance and implications of our findings.
>
> If you have any questions or need further clarification, please do not hesitate to let us know, and we will do our utmost to address them.
>
> > ***Q3:** Does I2EBench consider the aesthetic quality of image edits?*
> >
>
> **A3:**
>
> Thank you for your insightful comments. Aesthetic quality is indeed an important criterion in image editing. In response to your suggestion, we have integrated the Aesthetic Predictor’s Score (AP) to evaluate the aesthetic quality of edited images, similar to the approach used by InstructDiffusion [1]. Specifically, the AP score assesses the aesthetic quality of the generated images, employing a methodology akin to that used by LAION-5B [2], which utilizes the CLIP+MLP Aesthetic Score Predictor. A higher AP score indicates a better perceptual quality.
>
> We calculated the AP score for the edited images generated by different methods and averaged the scores for each evaluation dimension. This allows us to derive the AP scores for each method across various evaluation dimensions. As illustrated in **Tab. I and Tab. II of the rebuttal pdf**, our findings are twofold:
>
> - The difference in the Aesthetic Predictor’s Score between images edited using original instructions and those edited with diverse instructions is relatively small.
> - The variations in the Aesthetic Predictor’s Score across different dimensions are relatively large.
>
> These insights will enable us to refine our evaluation methodology and ensure a more comprehensive assessment of the aesthetic quality of image edits. Following your excellent suggestion, we will include the above experimental results and discussion in the revised manuscript.
>
> *[1] Instructdiffusion: A generalist modeling interface for vision tasks. CVPR. 2024.*
>
> *[2] Laion-5b: An open large-scale dataset for training next generation image-text models. NeurIPS. 2022.*

---

> > ### Comment · Reviewer_riYa · 2024-08-13
> >
> > Thanks for the response. I lean to keep my rating (5).

---

> > > ### Author Response · Authors · 2024-08-14
> > > **Response to Reviewer riYa**
> > >
> > > Thank you for your efforts in reviewing and for giving I2EBench a positive rating. Your insightful comments will significantly enhance the paper.

---

> ### Author Response · Authors · 2024-08-09
> **Sincere Request for Further Discussions**
>
> Dear Reviewer riYa,
>
> Thank you for your invaluable efforts and constructive feedback on our manuscript.
>
> As the discussion period draws to a close, we eagerly anticipate your thoughts on our response. We sincerely hope that our response meets your expectations. If there are any remaining concerns or aspects that require clarification, we are ready to address them as soon as possible.
>
> Best regards,
>
> The Authors

---

> > ### Comment · Area_Chair_71VD · 2024-08-13
> >
> > Dear Reviewer riYa:
> >
> > Thanks for reviewing this work. Would you mind to check authors' feedback and see if it resolves your concerns or you may have further comments?
> >
> > Best, AC

---

### Author Rebuttal · Authors · 2024-08-04

### Response To All Reviewers and Table data

We would like to extend our heartfelt gratitude to the reviewers for their valuable feedback and positive comments on our paper. Their insightful reviews have significantly enhanced the clarity and overall quality of our work.

We thank Reviewer riYa  $\color{red}{\mathbf{(5: Borderline Accept)}}$ for acknowledging the strengths of our paper. They highlighted the clear and easy-to-follow nature of our writing and recognized that I2EBench represents a significant advancement over previous works, benefiting the research community. They also appreciated our thoughtful inclusion of often overlooked low-level edits.

Reviewer 7DEf  $\color{red}{\mathbf{( 6: Weak Accept)}}$ praised our comprehensive explanation of the evaluation process and the wide range of evaluation dimensions covered by the benchmark. They also commended the valuable research insights provided, which can guide future model architecture design and data selection strategies. Additionally, they noted that the benchmark encompasses multiple types of image editing tasks.

Reviewer A2Kh $\color{red}{\mathbf{(7: Accept)}}$  recognized the remarkable comprehensiveness of I2EBench and its potential to significantly advance the field of instruction-based image editing. They appreciated the benchmark's wide array of evaluation dimensions and its strong emphasis on aligning with human perception.

Reviewer HHca  $\color{red}{\mathbf{(4: Borderline Reject)}}$ highlighted the novelty and efficacy of our proposed technical components. They acknowledged the importance of our work in addressing the lack of high-quality benchmarks in image editing. Furthermore, they found the evaluation pipeline reasonable and clear to follow, praised the innovative use of multimodal large language models for evaluating image editing results, and appreciated the overall clarity of our presentation.

We sincerely thank the reviewers for recognizing these strengths and for their positive feedback on the clarity, novelty, and effectiveness of our proposed methods. Their comments have further motivated us to address the concerns and improve upon the weaknesses pointed out in their reviews. We are dedicated to thoroughly addressing their concerns and providing a detailed response in our rebuttal. ***The tables involved in the rebuttal are all located in the PDF submitted during the rebuttal phase.***

---

### Decision · Program_Chairs · 2024-09-25

**Decision:**

Accept (poster)

**Comment:**

This paper was reviewed by four experts in the field.  The paper received mixed reviews: 5, 6, 8, 8. All reviewers find that the authors are addressing an important problem and the proposed evaluation datasets and metrics are important for the image editing community, and the analysis is thorough. AC also agrees that this is well-presented and important work and thus would suggest acceptance.

Still, this work remains some issues. Particularly, the usage of GPTv4 for supplementary evaluation requires further justification and the authors should also mention that as a potential limitation in the conclusion. Also, it shows a limited performance on low-level tasks, compared with high-level ones. Given that, we think there are still a lot of room for improvement and thus only recommended for poster presentation.

We strongly recommended the authors to carefully read all reviewers’ final feedback, and revise the manuscript as suggested in the final camera-ready version. We congratulate the authors on the acceptance of their paper!